# Water Uptake and Transport Properties of La_1−x_Ca_x_ScO_3−α_ Proton-Conducting Oxides

**DOI:** 10.3390/ma12142219

**Published:** 2019-07-10

**Authors:** Alyona Lesnichyova, Anna Stroeva, Semyon Belyakov, Andrey Farlenkov, Nikita Shevyrev, Maksim Plekhanov, Igor Khromushin, Tatyana Aksenova, Maxim Ananyev, Anton Kuzmin

**Affiliations:** 1Laboratory of Electrochemical Material Sciences, Institute of High-Temperature Electrochemistry, Yekaterinburg 620137, Russia; 2Institute of New Materials and Technologies, Ural Federal University, Yekaterinburg 620002, Russia; 3Institute of Chemical Engineering, Ural Federal University, Yekaterinburg 620002, Russia; 4Laboratory of Solid State Oxide Fuel Cells, Institute of High-Temperature Electrochemistry, Yekaterinburg 620137, Russia; 5Laboratory of Radiation Diffusion, Institute of Nuclear Physics, Almaty 050032, Kazakhstan

**Keywords:** LaScO_3_, defect structure, water uptake, proton conductivity and proton mobility

## Abstract

In this study, oxide materials La_1−x_Ca_x_ScO_3−α_ (x = 0.03, 0.05 and 0.10) were synthesized by the citric-nitrate combustion method. Single-phase solid solutions were obtained in the case of calcium content x = 0.03 and 0.05, whereas a calcium-enriched impurity phase was found at x = 0.10. Water uptake and release were studied by means of thermogravimetric analysis, thermodesorption spectroscopy and dilatometry. It was shown that lower calcium content in the main phase leads to a decrease in the water uptake. Conductivity was measured by four-probe direct current (DC) and two-probe ascension current (AC) methods at different temperatures, *p*O_2_ and *p*H_2_O. The effects of phase composition, microstructure and defect structure on electrical conductivity, as well as correlation between conductivity and water uptake experiments, were discussed. The contribution of ionic conductivity of La_1−x_Ca_x_ScO_3−α_ rises with decreasing temperature and increasing humidity. The domination of proton conductivity at temperatures below 500 °C under oxidizing and reducing atmospheres is exhibited. Water uptake and release as well as transport properties of La_1−x_Ca_x_ScO_3−α_ are compared with the properties of similar proton electrolytes, La_1−x_Sr_x_ScO_3−α_, and the possible reasons for their differences were discussed.

## 1. Introduction 

Oxide materials with proton conductivity have been attracting attention as key materials for highly efficient and environment friendly protonic ceramic fuel cells (PCFC). The use of proton-conducting electrolytes in comparison with oxygen-ion conductors has several advantages. For example, the proton transfer process has significantly lower activation energy than oxygen-ions, which leads to a reduction in the operating temperature of devices without any loss of their power characteristics [1,2,3]. 

Traditionally, researchers have focused on the study of solid oxides with A^2+^B^4+^O_3_-type perovskites (where A = Ba, Sr; and B = Ce, Zr) because of their high proton conductivity. However, some of these materials can decompose in atmospheres containing water vapor and carbon dioxide due to the high content of alkaline earth-metal cations [4,5,6].

Perovskites with A^3+^B^3+^O_3_ structure have a higher chemical stability and mechanical durability. Studies of the physicochemical properties of acceptor-doped La_1−x_M_x_BO_3−α_ (where M = Ca, Sr, Ba; and B = Y, Yb, Sc, In, Lu) solid solutions demonstrate a tendency to increase the total conductivity and proton mobility with the decrease in ionic radii of the B site cations [7,8,9]. Thus, oxide systems based on LaScO_3_ can be considered as promising materials for solid electrolytes of PCFC [3,4,5,6,7,8,9]. 

The transport properties of La_0.9_Sr_0.1_Sc_0.9_Mg_0.1_O_3−α_ [10,11]; LaSc_1−x_Mg_x_O_3−α_ [12], La_1−x_Sr_x_ScO_3−α_ [9,10,13,14,15,16] have been investigated in detail. Fujii et al. [12] reported the conductivities of La_1−x_Ca_x_ScO_3−α_. However, the influence of Sr–Ca substitution on water vapor solubility, conductivity and thermal expansion of La_1−x_Ca_x_ScO_3−α_, depending on temperature, partial pressures of oxygen (*p*O_2_) and water vapor (*p*H_2_O), is not described. The purpose of this study is to examine conductivity, defects concentration and ionic transport numbers of La_1−x_Ca_x_ScO_3−α_ (x = 0.03, 0.05, 0.1).

## 2. Materials and Methods

The La_0.97_Ca_0.03_ScO_3−α_ (LCS3), La_0.95_Ca_0.05_ScO_3−α_ (LCS5), La_0.9_Ca_0.1_ScO_3−α_ (LCS10) and La_0.9_Sr_0.1_ScO_3−α_ (LSS10) materials were prepared by the citric-nitrate combustion synthesis. The synthesis technique is given in detail in our previous work [16]. The stoichiometric amounts of La_2_O_3_, Sc_2_O_3_ (both precursors from “Plant of rare metals”, Novosibirsk, Russia) CaCO_3_ and SrCO_3_ (from OJSC «Acron», Veliky Novgorod, Russia) with purity higher than 99.5% were mixed with consideration of the calcination coefficients and then dissolved in a solution containing the required amounts of nitric acid. After the dissolution of all precursors was completed, citric acid was added, and the mixture evaporated prior to the combustion reaction. The obtained powders were calcined at 1200 °C for 2 h. The calcined powders were reground and pressed to obtain samples in a form of pellets and bars. These samples were sintered at 1650 °C for 5 h in air. The estimated density of the as-sintered samples was determined by the Archimedes method in kerosene. The theoretical density was determined from the ratio of volumes and molar mass of unit cells of the materials under study.

The phase composition and structure features of LCS powders were studied by the X-ray diffraction (XRD) analysis. XRD patterns were measured with a D/MAX-22(RIGAKU, Tokyo, Japan) diffractometer (CuKα radiation, λ = 1.5418 Å) at room temperature in the 2θ range 20–70° with a step of 0.02°. The phase composition of the samples was identified with the use of the PDF-2ICDD database. Crystal structure calculations were carried out using software MDIJade 6.5 (Livermore, USA) and Match-DEMO (Bonn, Germany). 

The surface and cross-section morphology of the ceramic samples was investigated by the scanning electron microscopy (SEM) on a TESCAN MIRA 3 LMU microscope (Brno, Czech Republic). Maps of elements distribution were obtained for the cross-sections of the ceramic samples by Energy-Dispersive X-ray (EDX) spectroscopy using Oxford Instruments (Abingdon, UK) INCA Energy 350 X-Max 80 equipment. The preparation procedure for the SEM characterization of the samples cross sections included an epoxy impregnation in rough vacuum (residual pressure 10^−2^ atmospheres), grinding and polishing using Allied MetPrep 4/PH-4 (Compton, CA, USA) System grinding and polishing machine, as well as diamond suspensions.

Studies of the release of H_2_O and CO_2_ from LCS and LSS were performed by thermodesorption spectroscopy (TDS) using an equipment based on vacuum chamber, heating device and radio frequency mass-spectrometer, which are described in detail in [17,18]. The vacuum during the experiment was 10^−5^ Pa and the heating rate was 42 °C min^−1^. The water saturation of the dense ceramic samples was carried out by annealing at 500 °C and *p*H_2_O = 4.5 kPa during 15 h.

The thermal expansion of the dense ceramic bars was studied using a quartz dilatometer and TESA Tesatronic TT-80 (Renens, Switzerland) equipment from room temperature to 900 °C with a heating/cooling rate of 2 °C min^−1^ in dry (*p*H_2_O < 0.1 kPa) and wet (*p*H_2_O = 3.2 kPa) air atmospheres.

The thermogravimetric analysis (TGA) was performed on the powder samples with specific surface area (S_sp_) of 1.1 ± 0.2 m^2^/g using a Netzsch STA Jupiter 449 F3 (Selb, Germany) analyzer with water vapor generator Adrop Asteam DV2MK. Powders specific surface area for TGA measurements was determined by Brunauer-Emmett-Teller method using Sorbi N.4.1 (Meta, Russia). The as prepared powders were heated up to 950 °C and held at this temperature for 8 h under the dry argon (*p*H_2_O < 10^2^ Pa, 99.9998% of purity). Afterwards the carrier gas was saturated with water vapor (*p*H_2_O = 24.3 kPa) and then the increase in weight was recorded upon cooling from 950 °C to 300 °C with a cooling rate of 30 °C h^−1^ and exposure at every 100 °C during 2 h. The proton concentration was obtained from the weight change during TGA as follows:(1)$$\Delta n_{OH}=\frac{2\cdot \Delta m\cdot M_{oxide}}{m_{oxide}\cdot M_{H_{2}O}}$$
where *M_oxide_* and *M_H_2_O_* are molecular weights of the oxides and water, *Δm* and *m_oxide_* are weight change and weight of the specimen, respectively.

The electrical conductivity was measured as a function of temperature (T = 900–300 °C), oxygen partial pressure (*p*O_2_ = 2.1 × 10^4^–10^−18^ Pa) and water partial pressure (*p*H_2_O < 0.1 kPa and 4.5 kPa) by both two-probe impedance and four-probe DC methods. Porous platinum electrodes were fabricated by firing in air at 1000 °C for 1 h. Electrochemical impedance spectroscopy (EIS) was used to estimate the contributions of bulk and grain boundary resistances to the total resistance of ceramic the samples under study. The complex impedance was measured in the frequency range from 100 mHz to 8 MHz using electrochemical analyzer IM6 (Zahner Elektrik, Kronach, Germany). At temperatures above 600 °C, the impedance spectra are shifted towards higher frequencies, so the semi-circle corresponding to the bulk resistances become indistinguishable. The impedance spectra were analyzed using an equivalent circuit model (EC) (Figure 1, insert 1). In the case of EC, R is the resistance of the simulated electrochemical process, CPE is a constant phase element; and indices 1 and 2 indices are the corresponding bulk and grain boundary processes, respectively [19]. The accuracy of equivalent circuit model was validated by additional interpretation of EIS data using the distribution of relaxation times (DRT) method. DRT analysis was carried out using software developed by the authors of [20]. The temperature dependences of conductivity for LCS5 sample, obtained by EC and DRT methods, have a high convergence (Figure 1).

## 3. Results and Discussion

### 3.1. Ceramics Characterization

The XRD analysis demonstrated that LCS3, LCS5 and LCS10 samples are single-phase. All diffraction peaks in each pattern were indexed assuming a LaScO_3_-type orthorhombic distorted perovskite-type unit cell (Figure 2). The corresponding data on unit cell volumes are listed in Table 1. The unit cell volume rises slightly with Ca^2+^ concentration increasing in LCS. This means that the effective volume of [La_2_O_3_] group in perovskite crystal lattice is slightly less (≈ 5%) than the volume of [Ca_2_O_2_($V_{O}^{••}$)] group, which may be due to the larger effective radius of the oxygen vacancy ($V_{O}^{••}$) than the oxygen ion. Another explanation could be ralated to the water incorporation into a crystal lattice [21], which is relevant, considering that powders were not calcined in dry air prior to XRD analysis.

The relative density of LCS ceramic samples reaches 97%–98% (Table 1). To assess the morphological characteristics of the sintered ceramics, the SEM study of surface and cross-section was carried out. It can be seen that all of the materials under study have nonporous microstructure with close-packed and well-crystalline grains (Figure 3a). The microstructure of LCS3 sample is fine-grained, and the grain size varies within a range of 0.1–2.0 μm, while LCS5 has an average grain size about 2 μm. SEM studies have shown that the LCS10 sample contains calcium-enriched impurity phases with particle sizes of 0.5–2 μm, despite the fact that this sample is conditionally single-phase according to XRD. An EDX analysis of the cross sections of the LCS ceramics confirmed the non-uniform distribution of calcium in the LCS10 sample (Figure 3b). It is noteworthy that La_1−x_Ca_x_ScO_3−α_ (x = 0.03–0.05) samples have uniform distribution of all elements, as well as La_1−x_Sr_x_ScO_3−α_ (x = 0.05–0.1) [16]. According to the quantitative element analysis (Table 2), cations ratio in A and B sites corresponds to the sample’s nominal composition. The relative content of Ca in LCS3 and LCS5 is a little less than specified (2.3 at.% and 4.4 at.% respectively). But in the case of LCS10, calcium content was significantly smaller (~8 at.%), which most likely corresponds to an unequal distribution of Ca between two phases. No other impurities were detected within the detection limit.

Thus, according to our data the solubility limit Ca^2+^ into La^3+^ is located within the range of x = 0.05–0.1. At Ca concentration of x = 0.1 second phase with higher Ca content could be observed. On the other hand, Fujji et al. [12] presented La_1−x_Ca_x_ScO_3−α_ (x = 0.05–0.2) as single-phase according only XRD data. Unfortunately, no further details were provided. 

### 3.2. Thermodesorption Spectroscopy

It is well-known that oxygen vacancies induce incorporation of water molecules and formation of OH-defects:(2)$$H_{2}O+V_{O}^{••}+O_{O}^{\times }=2OH_{O}^{•}$$
where the Kröger-Vink notation is used to describe an oxygen vacancy $V_{O}^{••}$, oxygen ions at an oxygen lattice sites $O_{O}^{X}$ and OH-defects on oxygen sites $OH_{O}^{•}$ [22,23].

The content of oxygen vacancies in La_1−x_Ca_x_ScO_3−α_ is determined by the concentration of the Ca acceptor dopant, which formed according to the following equation:(3)$$CaO(-LaO_{1.5})\to Ca_{La}^{/}+\frac{1}{2}V_{O}^{••}+O_{O}^{\times }$$
where $Ca_{La}^{/}$ are atoms of Ca in a La-sites of the crystal lattice.

Results of the TDS study of water release from hydrated LCS and LSS are presented in Figure 4a. The intensity of water release of LCS10 is almost twice as high as for LCS5, which mostly related to the difference in oxygen vacancies content (Equations (2) and (3)). In the case of LCS5, the maximum of the TD-spectrum shifts to higher temperatures. With a similar doping level of strontium in LSS10, the release of water is much more intense and begins with lower temperatures. Spectra of LCS10 and LSS10 may differ due to presence of second phase in LCS10, that is, lower Ca content in the main phase, hence reduced concentration of oxygen vacancies. On the other hand, this may be caused by the kinetic features of the water release. Since the TDS measurements were carried out on dense ceramic samples, the kinetic factor could be decisive, given the high heating rate of 42 °C min^−1^. 

The CO_2_ release is observed only in LCS10 (Figure 4b) and was most likely caused by the impurity content that was detected in SEM (Figure 3). It is likely that the calcium-enriched impurity phases initiated the chemical interaction with CO_2_. This may have caused kinetic difficulty of water exchanging and decreased its release on TD-spectrum. No evidence of CO_2_ release was detected in LCS5, as well as LSS10. This indicates a good chemical stability of materials in CO_2_-containing atmospheres.

### 3.3. Thermal Expansion

The temperature behavior of dilatometric curves for LCS ceramic samples in the heating regime in dry (*p*H_2_O < 0.1 kPa) and wet (*p*H_2_O = 4.2 kPa) air is nearly linear and monotonic (Figure 5a). This confirms the absence of phase transitions in the examined temperature range. Average TEC values in dry air are close to each other and equal to (8.55 ± 0.03) × 10^−6^ K^−1^ and (8.52 ± 0.03) × 10^−6^ K^−1^ for LCS5 and LCS10, respectively. In wet air conditions, TEC slightly decreases for LCS5 to (8.49 ± 0.03) × 10^−6^ K^−1^ and increases for LCS10 to (8.68 ± 0.03) × 10^−6^ K^−1^. Obtained values of TEC for LCS are close to those for LSS electrolytes [13].

The thermal expansion of hydrated LCS5 and LCS10 samples in dry air (Figure 5a) demonstrates the impact of water release on the crystal lattice of the materials during the temperature increasing. Figure 5b,c shows the difference between dilatometric curves of hydrated and dried LCS ceramic samples in comparison with TDS results. The obtained difference is in good agreement with TDS data, where water release above 600 °C can be observed. Earlier, we showed similar changes on dilatometric curves of LSS ceramics [13]. Thus, the water release in lanthanum scandates is characteristic above 600 °C; however, in the case of Ca-doped materials it is less intense. In Figure 5a we displayed the dilatometric curve for undoped LaScO_3_, obtained by the authors of [24]. Measurements were carried out for both dried and humidified samples in dry air. Unlike LSS and LCS samples undoped LaScO_3_ has no inflections on the linear expansion temperature dependence, which means water was not incorporated into crystal lattice during hydration. Nevertheless, the TEC value of LaScO_3_ is quite close to those of LSS and LCS and is (8.74 ± 0.03) × 10^−6^ K^−1^ at 500 °C [24].

### 3.4. Water Uptake

Figure 6a shows the weight change and hydration isobars under water partial pressure *p*H_2_O = 24.3 kPa of the single-phase LCS5 powder (S_sp_**=** 1.1 ± 0.2 m^2^/g), obtained by trituration of ceramic sample.

According to Equation (2), the equilibrium constant *K*_H_ of hydration reaction is expressed as
(4)$$K_{H}=\frac{{\left[OH_{O}^{•}\right]}^{2}}{[V_{O}^{••}][O_{O}^{\times }]pH_{2}O}$$
where $[OH_{O}^{•}]=\Delta n_{OH}$, $\left[V_{O}^{••}\right]$ and $\left[O_{O}^{\times }\right]$ are concentrations of OH-defects, oxygen vacancies and lattice oxygen, respectively. In order to calculate *K*_H_, the following electroneutrality condition and site restriction for proton conducting perovskite oxide are employed by neglecting concentration of electronic holes:(5)$$2\left[V_{O}^{••}\right]+\left[OH_{O}^{•}\right]=\left[Ca_{La}^{/}\right]$$
(6)$$[O_{O}^{\times }]=3-[V_{O}^{••}]-[OH_{O}^{•}],$$
where $\left[Ca_{La}^{/}\right]$ is the concentration of acceptor impurity *x* in LCS5 oxide. Equilibrium constant *K*_H_ (4) of hydration reaction (2) can be described in terms of enthalpy Δ*H_hydr_* and entropy Δ*S_hydr_*:(7)$$K_{H}=\exp\left(\frac{\Delta S_{hydr}}{R}\right)\exp\left(-\frac{\Delta H_{hydr}}{RT}\right),$$
where *R* is the universal gas constant and *T* is the absolute temperature. The standard enthalpy Δ*H_hydr_* and entropy Δ*S_hydr_* were −132 ± 5 kJ mol^–1^ and −126 ± 5 J mol^−1^ K^−1^, respectively. The obtained values are comparable with the same values for the best-known perovskite-like oxides with orthorhombic type structure such us BaCe_1−*x*_Y*_x_*O_3−α_ [23], La_1−*x*_Sr*_x_*YbO_3−α_ [25] and La_1−*x*_Sr*_x_*ScO_3−α_ [21,26].

According to Equations (2) and (3), the value of maximal water uptake (saturation level) has to be equal to the acceptor doping concentration: $n_{OH}$ = 0.05 for investigated LCS5 sample. In the case of LCS5, we do not observe exact match between the saturation level $n_{OH}$ and the acceptor concentration *x* unlike La_1−*x*_Sr*_x_*ScO_3−α_ [21], Figure 6b. The difference between the effective and nominal dopant concentrations can be partly caused by reduced calcium contant in comparison with the specified composition (Section 3.1, Table 2). On the other hand, this phenomenon is often explained by the existence of a strong bond of the acceptor defects with oxygen vacancies, which do not participate in the hydration process [25]. In the case of LaScO_3_-based oxides, this phenomenon can be associated with not fully filled oxygen vacancies due to non-equality of O_1_ and O_2_ oxygen positions [26,27].

### 3.5. Transport Properties

Acceptor-doped materials based on the LaScO_3_ are mixed ion-hole conductors in oxidizing conditions [9,10,11,12,13,14]. Proton and oxygen-ion defects are formed according to Equations (2) and (3), respectively. The formation of holes (h^•^) occurs at interaction atmospheric oxygen with oxygen vacancies:(8)$$\frac{1}{2}O_{2}+V_{O}^{••}=2h^{•}+O_{O}^{\times },$$

The concentration of electronic holes is equal to
(9)$$p=\frac{K_{8}^{1/2}{\left[V_{O}^{••}\right]}^{1/2}}{{\left[O_{O}^{\times }\right]}^{1/2}}pO_{2}^{1/4}$$
where K_8_ is the equilibrium constant of Equation (8).

Thus, the general condition of electroneutrality for La_1−*x*_Ca*_x_*ScO_3−α_ in wet air is
(10)$$2[V_{O}^{••}]+[OH_{O}^{•}]+p=[Ca_{La}^{/}].$$

In reducing atmospheres, these materials are ionic conductors [9,11,14]; therefore, the concentration of holes and electrons is neglectable and the electroneutrality condition can be transformed to
(11)$$2[V_{O}^{••}]+[OH_{O}^{•}]=[Ca_{La}^{/}].$$

Figure 7 shows Arrhenius plots of electrical conductivities of LCS under wet air (*p*H_2_O = 4.5 kPa). The total conductivity of LCS5 sample even exceeds LCS10 conductivity (Figure 7a). This may be due to the presence in the LCS10 sample of calcium-enriched impurity phases (Figure 3), which adversely affect the transport properties of the material. The LCS3 sample has the lowest conductivity. As illustrated in Figure 7b, with decrease in temperature, the difference between the grain boundary and bulk conductivities increases resulting in the domination of the grain boundary resistivity, which determines the total conductivity. Such behavior is tipical for proton conducting electrolytes [28,29], and often explained by impurities segregation or by chemical interaction with gas phase resulting in by-products formation on grain boundaries. In the case of LCS3 the grain boundary (GB) conductivity remains significantly lower than the bulk one. This effect probably results from the fine-grained microstructure of LCS3 (Figure 3), and thus the greater grain boundary length. The small difference between total conductivities of LCS5 and LCS10 (Figure 7a) was caused by the difference of their bulk and GB contributions (Figure 7b). The difference in their GB conductivity decreases with increasing temperature, and the difference in bulk conductivity, on the contrary, increases.

The values of *E_act_* of total conductivity for LCS are summarized in Table 3. LCS3 has the highest *E_ac_*_t_ at all measurement conditions. At the same time, the values of activation energies for LCS5 and LCS10 are comparable in a similar enviroment. A decrease in temperature leads to an increase in *E_act_* of LCS, which may be due to the influence of grain boundaries at low temperatures (Figure 7b).

Figure 7a also presents electrical conductivities of LCS5 and LCS10 in wet air in Arrhenius plot from the paper of Fujji et al. [12]. It can be seen that the conductivity values of LCS5 and LCS10 obtained in the present study are higher by an order of magnitude than in [12]. This may be due to the microstructure features (grain sizes, relative density and others) of the samples, the details of which are not described by Fujji et al. The difference in the electrical conductivity of LSS10 and LCS (5 and 10) becomes apparent only in the low temperature range (Figure 7a), since LSS10 has higher bulk and GB conductivities (Figure 7b).

The total conductivity of LCS was previously determined by microstructure features and the resistance of GB. At the same time, the bulk properties are mostly defined by defect structure, which can be illustrated by measurements in different oxygen partial pressures (*p*O_2_). The total conductivity of LCS samples was measured depending on *p*O_2_ at 700 °C under wet (*p*H_2_O = 4.5 kPa) conditions, as shown in Figure 8. As the *p*O_2_ decreases, the total conductivity of all these samples decreases and reaches a constant value that corresponds to the ionic conductivity. The evaluated ion transfer numbers for LCS3, LCS5 and LCS10 are 0.45, 0.32 and 0.37, respectively. The value of hole conductivity was calculated as the difference between the total and ionic conductivities (Figure 8). The slope of the hole conductivity for all samples agrees with the theoretical value of 1/4 according to Equation (9).

According to Equation (9), the hole and proton concentration increases along with the concentration of oxygen vacancies and according to Equation (3), that is, in the order of LCS3–LCS5–LCS10. However, it was found that the hole and ionic conductivity in LCS5 is slightly higher than that of LCS10, despite the lower concentration of the Ca acceptor dopant. These experimental facts could be explained by reduced calcium content or by a decrease in the charge carriers mobility due to the presence of an impurity phase or the formation of associate defect complex ($Ca_{La}^{/}$-$V_{O}^{••}$-$Ca_{La}^{/}$) [30]. 

The influence of humidity and *pO_2_* on total conductivity is considered in the example of the LCS5 sample (Figure 9a). At temperatures above 600 °C, the total conductivity of LCS5 is comparable in both dry (*p*H_2_O < 0.1 kPa) and wet (*p*H_2_O = 4.5 kPa) air, and even exceeds the latter. This behavior is explained by the domination of conductivity of electronic holes, which decreases with increasing humidity and decreasing temperature (Figure 9b). In dry air, the ion transfer numbers are lower than 0.1 at 800 °C and 700 °C. These values increase to 0.17 and 0.32, when the atmosphere is humidified.

The total conductivity of LCS5 in a wet (*p*H_2_O = 4.5 kPa) reducing atmosphere does not contain the contribution of hole conductivity (Figure 9a). Therefore, at high temperatures (600–900 °C) the conductivity of the samples in wet reducing atmosphere is significantly lower than that in wet air, while in the temperature range of 500–600 °C this difference becomes insignificant. The protonic conductivity is dominant in both oxidizing and reducing atmospheres. 

As can be seen in Table 3, at low temperatures LCS5 effective activation energy decreases along with humidity and oxygen partial pressure due to an increase in the proton conductivity contribution. At high temperatures, the lowest value of activation energy is observed in the reducing atmosphere. The difference between dried and humidified air results from complex relationship of hole, oxygen ion and proton conductivity contributions. Finally, the activation energies of total conductivity of LSS electrolytes is lower compared than that in LCS materials under all experimental conditions, which suggests the fact that Sr-doped materials are characterized by higher proton conductivity.

In this work, long-term durability has been also studied. It is evident that the electrical conductivity of the studied materials remains constant for 120 h at 700 °C in an oxidizing wet atmosphere (*p*H_2_O = 4.5 kPa) (Figure 10). Therefore, these materials are stable enough under suggested operating temperatures and can be considered as promising proton-conducting electrolytes for electrochemical devices.

## 4. Conclusions

In the present work, phase composition, microstructure features, thermal expansion behavior, processes of water uptake and release and electrical conductivity of La_1−x_Ca_x_ScO_3−α_ (x = 0.03, 0.05 and 0.10) materials were investigated.

The La_1−x_Ca_x_ScO_3−α_ (x = 0.03, 0.05 and 0.1) dense ceramic materials were synthesized by the citrate-nitrate combustion technique. An X-ray diffraction analysis demonstrated that samples under study are single-phase, whereas according to the SEM study LCS10 sample contains a Ca-enriched impurity together with the main perovskite phase. The presence of the impurity leads to lower calcium content in the main phase and a decrease in water release on TD-spectrum, compared with La_0.9_Sr_0.1_ScO_3−α_. This also results in an interaction with CO_2_. No evidence of CO_2_ release was detected in LCS5 and LSS10, indicating good chemical stability of these materials in CO_2_-containing atmospheres. 

The concentration of the dopant and the composition of the gas phase have a slight effect on the thermal linear expansion coefficient of the LCS5 and LCS10 samples. For the hydrated samples, observed inflections on the linear expansion temperature dependencies are in good agreement with those obtained by TDS temperature at which desorption begins. Proton concentrations, obtained by TGA, for LCS5 does not reach dopant level. 

The LCS materials under study have mixed ion-hole conductivity in wet air. The higher total and bulk conductivities are obtained for LCS5 samples. LCS3 grain boundary conductivity is up to two orders of magnitude lower compared with other samples. The hole conductivity of LCS5 is slightly higher than that of LCS10, despite the lower concentration of the Ca acceptor dopant. The decrease in the electrical conductivity of LCS10 is possibly due to presence of Ca-enriched impurity phases or formation of associated defects.

The contribution of ionic conductivity of LCS rises with decreasing temperature and increasing humidity of the atmosphere. Under reducing atmospheres, LCS materials are pure ionic conductors. At temperatures below 500 °C, the protonic conductivity is dominant in all atmospheres. The results obtained here allow the LCS5 to be considered as the promising proton-conducting electrolyte for electrochemical devices.

## Figures and Tables

**Figure 1 materials-12-02219-f001:**
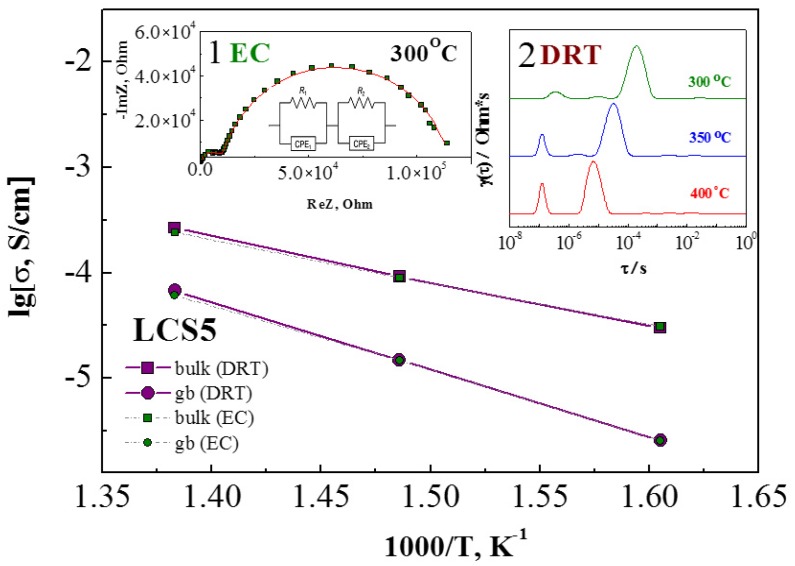
The temperature dependences of conductivity for LCS5 sample, obtained after interpretation by two independent methods, such as EC—equivalent circuit (insert 1), and DRT—relaxation time distribution (insert 2).

**Figure 2 materials-12-02219-f002:**
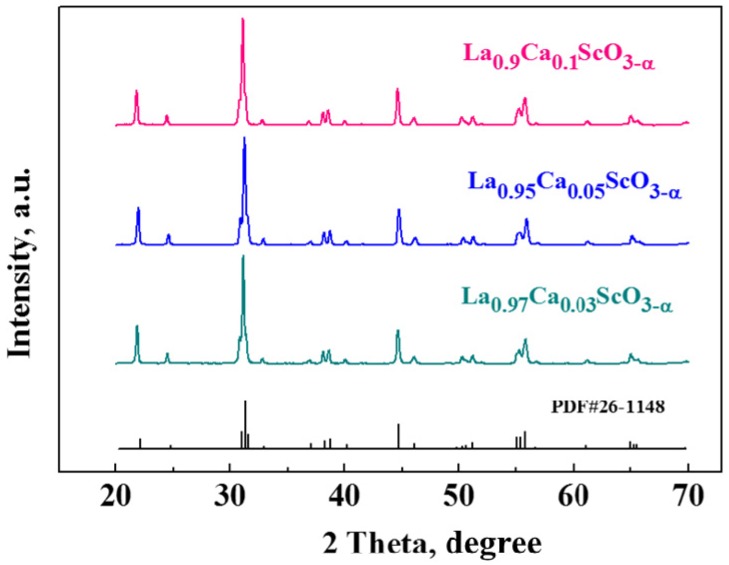
XRD patterns of LCS3, LCS5 and LCS10.

**Figure 3 materials-12-02219-f003:**
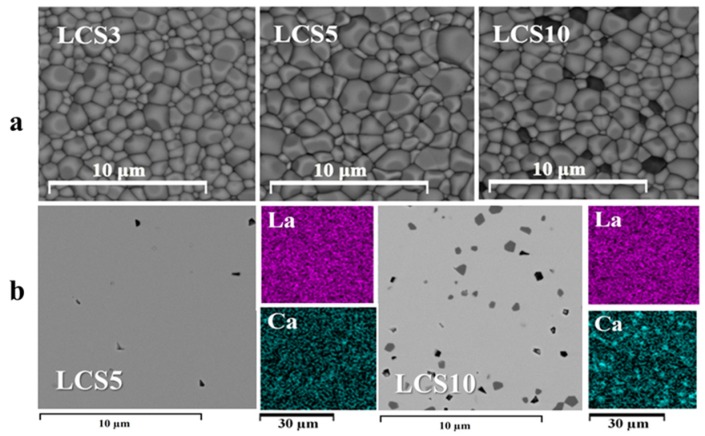
SEM images (BSE) of surface morphology: (**a**) With cross-sections with EDX maps (both La and Ca distribution); and (**b**) for LCS ceramic samples.

**Figure 4 materials-12-02219-f004:**
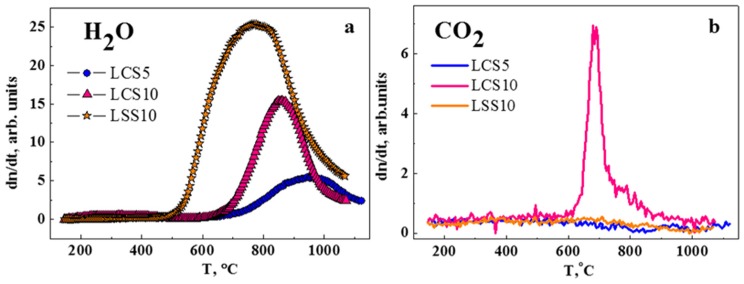
TDS results for LCS5, LCS10 and LSS10 ceramic samples: (**a**) Release of H_2_O; (**b**) release of CO_2_.

**Figure 5 materials-12-02219-f005:**
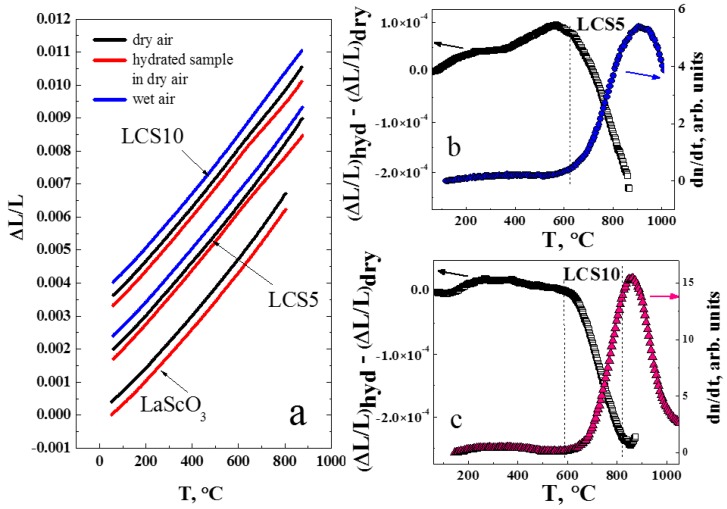
Thermal expansion measurements of LCS and LaScO_3_ [24] ceramic samples: (**a**) Initial dilatometric curves in dry (*p*H_2_O < 0.1 kPa) and wet (*p*H_2_O = 3.2 kPa) air; (**b**) the difference between expansion of hydrated and dried LCS5 (left axe) and TDS of water (right axe); (**c**) the difference between expansion of hydrated and dried LCS10 (left axe) and TDS of water (right axe).

**Figure 6 materials-12-02219-f006:**
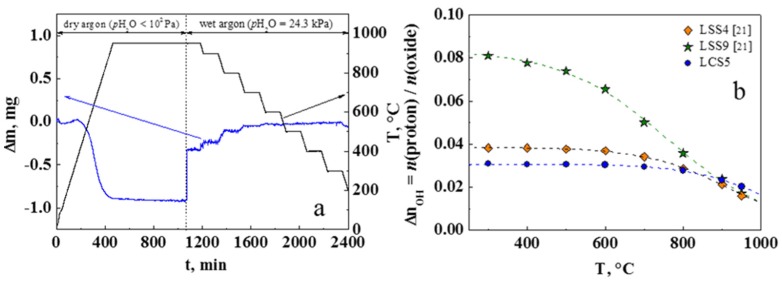
TGA results for lanthanum scandates: (**a**) Weight change of the LCS5 during hydration; (**b**) the temperature dependence of the proton concentration for LCS5 and comparison with literature data for strontium doped lanthanum scandates [21], *p*H_2_O = 24.3 kPa. Symbols correspond to the concentrations obtained from the TGA data and dashed lines are the fitting curve.

**Figure 7 materials-12-02219-f007:**
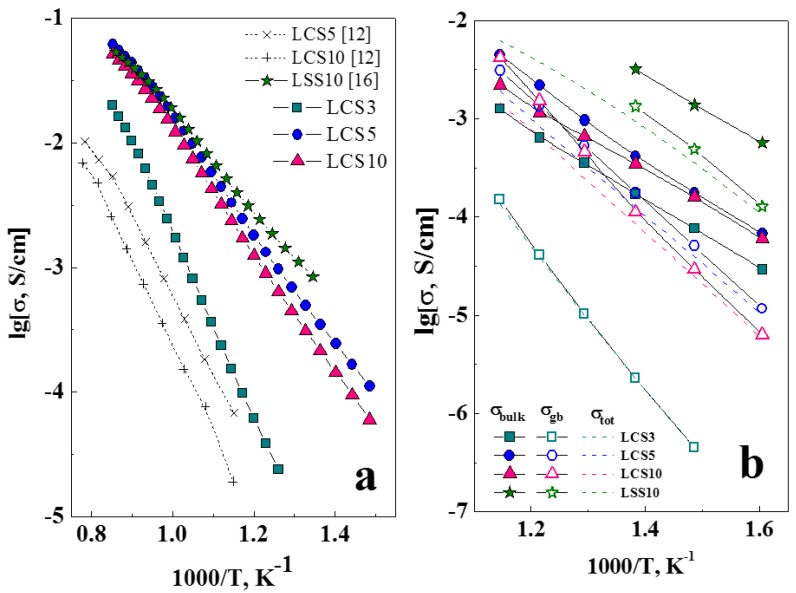
Temperature dependencies of electrical conductivities for LCS3, LCS5, LCS10 ceramics under wet air (*p*H_2_O = 4.5 kPa) by four-probe DC method (**a**), and by two-probe impedance method (**b**).

**Figure 8 materials-12-02219-f008:**
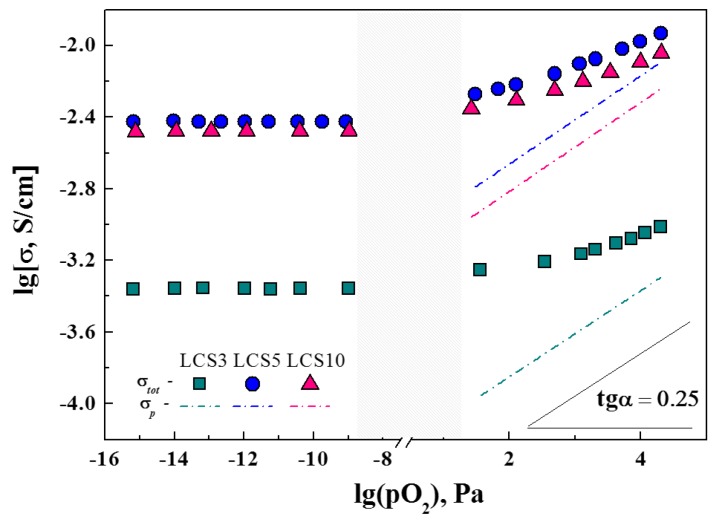
Dependences of conductivity of LCS samples on the oxygen partial pressure at 700 °C and *p*H_2_O = 4.5 kPa. Dashed lines are the hole conductivity of LCS.

**Figure 9 materials-12-02219-f009:**
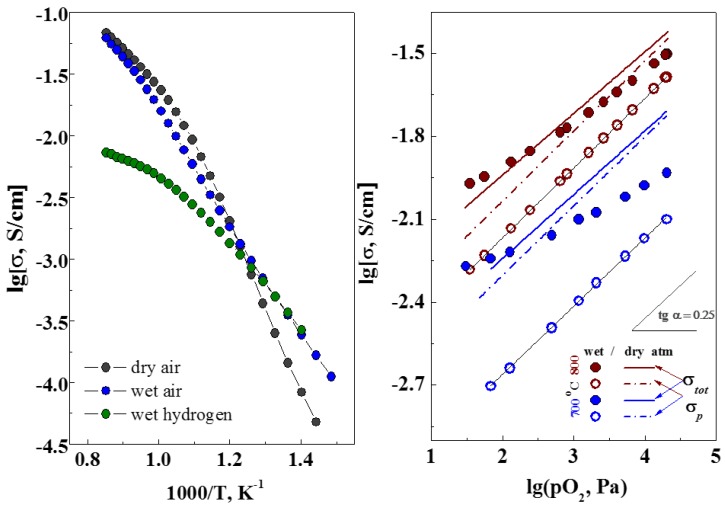
Conductivity dependences on the temperature (**a**) and on the oxygen partial pressure (**b**) for the LCS5 sample ceramics in dry (*p*H_2_O < 0.1 kPa) and wet (*p*H_2_O = 4.5 kPa) air, as well as in a wet (*p*H_2_O = 4.5 kPa) reducing atmosphere.

**Figure 10 materials-12-02219-f010:**
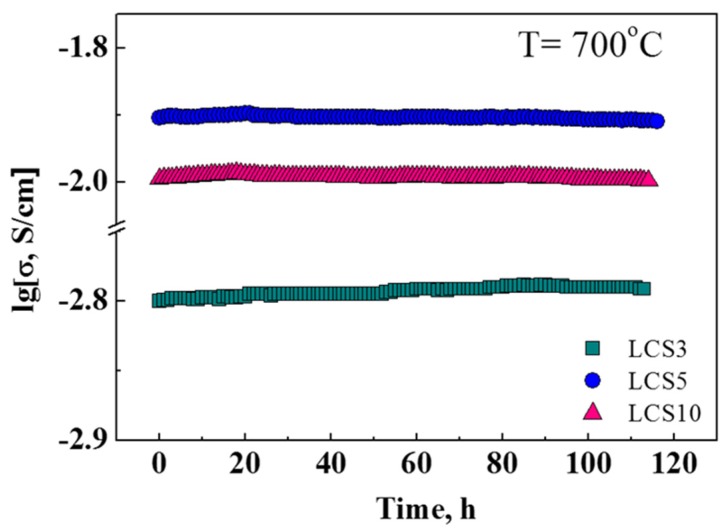
Evolution of electrical conductivity at 700 °C for 120 h in an oxidizing atmosphere (*p*H_2_O = 4.5 kPa).

**Table 1 materials-12-02219-t001:** Unit cell volume and density of ceramic samples.

Sample	V, Å^3^	Estimated Density, g/cm^3^	Relative Density, %
LCS3	265.83	5.58 ± 0.2	97
LCS5	266.12	5.49 ± 0.2	97
LCS10	267.88	5.45 ± 0.2	98

**Table 2 materials-12-02219-t002:** Results of EDX analysis of LCS3, LCS5, LCS10 ceramic samples (in relative at. %).

Element	La_0.97_Ca_0.03_ScO_3−α_	La_0.95_Ca_0.05_ScO_3−α_	La_0.90_Ca_0.10_ScO_3−α_
La	49.1 ± 0.4	47.8 ± 0.4	45.7 ± 0.3
Ca	1.2 ± 0.1	2.2 ± 0.1	4.0 ± 0.2
Sc	49.7 ± 0.4	50.0 ± 0.5	50.3 ± 0.5

**Table 3 materials-12-02219-t003:** Effective activation energies of the total conductivity of the LCS and LSS materials in various atmospheres. HT and LT are relatively linear high temperature (750–900 °C) and low temperature (500–650 °C) regions.

Sample	Effective Activation Energies, eV					
	Air				Hydrogen	
	*p*H_2_O = 4.5 kPa		*p*H_2_O < 0.1 kPa		*p*H_2_O = 4.5 kPa	
	HT	LT	HT	LT	HT	LT
LCS3	1.44	1.54	1.34	1.85	0.82	1.17
LCS5	0.87	0.96	0.72	1.39	0.33	0.70
LCS10	0.91	1.02	0.76	1.45	0.34	0.78
LSS5 [12]	0.69	0.79	0.66	0.92	0.33	0.53
LSS10 [12]	0.63	0.69	0.67	0.72	0.39	0.44

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
