# Peer review of "Water Uptake and Transport Properties of La1−xCaxScO3−α Proton-Conducting Oxides"

_materials, 2019, doi:10.3390/ma12142219_

Reviewer 1 Report

1)     Please indicate where you bought the precursors from and individual purity content. Also, citric acid content vs the total metal nitrates as well as combustion temperature prior to sintering should be indicated. Any detail for future comparisons will help you and other groups.

2)     Were there any grinding and polishing steps for the pellets and bars? The cross section SEM image in Fig. 3 doesn’t show any grains. It seems to be a polished sample. Please give details.

3)     Unit cell volume in Table 3: Give detail on how you obtained this information. Did you use a fitting program?

4)     The oxygen content in EDX analysis can be erroneous. Just give relative at. % in Table 2 in order to have a comparison for intended stoichiometry for La, Ca and Sc.

5)     The creation of pdf document resulted in a mix up in Eq 3. Please make sure, the print is corrrect for all eqs.

6)     As I understand, the authors studied LSS10 previously. As this sample was compared in many figures to LCS samples, the authors should give as much details on LSS in the introduction and the experimental part.

7)     Line 190: I don’t understand what the authors meant by inflection point for LS sample. Is it related to experiments in wet air?

8)     Line 188. has a lower intense ..? Reformulate

9)     Electroneutrality doesn’t hold in Eq. 5, 10, 11.

10)  Eq 6: Can you explicitely write where the 3 comes from?

11)  Eq. 8 and 9. Is p equal to 2h.? Minus signs are missing in Eq 9.

12)  Table 3 is missing the unit eV.

13)  Line 120: analysis

14)  Line 247: is comparable to,.. even exceeds. Simply write exceeds.

Reviewer 2 Report

High-temperature proton conductors, as electrolytes, in Protonic Ceramic Fuel Cell (PCFC) technology characterized by a high ionic conductivity at the intermediate temperatures may surpass one of commonly used oxide ion conductors. Authors have synthesized the perovskite material using combustion technique and at the optimal level of calcium single phase was reported. Comprehensive results of structural, thermogravimetric, and transport properties of proton-conducting material are presented. The presented work appears to be original (part of the work may be presented at the conference with a similar title) and the work is publishable after moderate revisions.

My specific comments are below.

·         Part of the work has been submitted at the conference, please acknowledge it appropriately.

·         Both the sections Abstract and section 2, the opening sentence is about “prepared by the citric-nitrate combustion synthesis” but no provided no further details. Please provide the temperature, fuel, and precursors used. Combustion technique needs to be referred back to the literature such as Chem Engg Journal 253 (2014) 502).

·         The first reports on proton-conducting ceramics were presented by Iwahara and co-workers in the 1980s, which has not been clearly mentioned.

·         Arguably, materials like BaCeO3 provide the highest value of proton conductivity then why should not one focus on those rather LaCaScO3? Please justify.

·         Were the precursor La2O3 preheated up to 800 °C (before use) in order to remove water and ensure proper stoichiometry. There is no mention about that in Section 2.

·         In the impedance spectroscopy analysis, please provide the values of Rs and Rct. “The semi-circle corresponding to the bulk resistances” and the EC model need to be referred back to the literature Electrochim. Acta 137 (2014) 497; and Industrial and Engineering Chemistry Research 53(39) (2014) 14993).

·         In the XRD pattern, impurity peaks of Ca be marked/identified.

·         Does the equation [2] corresponds to decoupled chemical diffusion of hydrogen and oxygen in the electrode? It is unclear though.

·         Please explain the acronym of TDS?

·         What is the average crystallite size?

·         Are these materials strongly reacting with moisture from the air?

·         What parameters influence the ionic conductivity (proton content or electron-hole at different temp)?

·         Please state the values of the activation energy of transference numbers.

·         Conclusions need to be significantly condensed, just focussing on the results.
